# Carbonate Lake Sediments in the Plastics Processing-Preliminary Polylactide Composite Case Study: Mechanical and Structural Properties

**DOI:** 10.3390/ma15176106

**Published:** 2022-09-02

**Authors:** Grzegorz Borkowski, Agnieszka Martyła, Marta Dobrosielska, Piotr Marciniak, Ewa Gabriel, Julia Głowacka, Marek Jałbrzykowski, Daria Pakuła, Robert E. Przekop

**Affiliations:** 1Faculty of Geographical and Geological Sciences, Adam Mickiewicz University, ul. B. Krygowskiego 10, 61-680 Poznan, Poland; 2Centre for Advanced Technologies, Adam Mickiewicz University in Poznań, ul. Uniwersytetu Poznańskiego 10, 61-614 Poznan, Poland; 3Faculty of Materials Science and Engineering, Warsaw University of Technology, ul. Wołoska 141, 02-507 Warsaw, Poland; 4Faculty of Chemistry, Adam Mickiewicz University in Poznań, ul. Uniwersytetu Poznańskiego 8, 61-614 Poznan, Poland; 5Faculty of Mechanical Engineering, Bialystok University of Technology, ul. Wiejska 45c, 15-351 Bialystok, Poland

**Keywords:** PLA, polylactide, composite, carbonates, lake sediments, natural filler, calcium carbonate

## Abstract

In this study, the influence of carbonate lake sediments (Polylactide/Carbonate Lake Sediments–PLA/CLS) on the mechanical and structural properties of polylactide matrix composites was investigated. Two fractions of sediments originating from 3–8 and 8–12 m were analysed for differences in particle size by distribution (Dynamic Light Scattering–DLS), phase composition (X-ray Diffraction–XRD), the presence of surface functional groups (Fourier Transform-Infrared–FT-IR), and thermal stability (Thermogravimetric Analysis–TGA). Microscopic observations of the composite fractures were also performed. The effect of the precipitate fraction on the mechanical properties of the composites before and after conditioning in the weathering chamber was verified through peel strength, flexural strength, and impact strength tests. A melt flow rate study was performed to evaluate the effect of sediment on the processing properties of the PLA/CLS composite. Hydrophobic-hydrophilic properties were also investigated, and fracture analysis was performed by optical and electron microscopy. The addition of carbon lake sediments to PLA allows for the obtention of composites resistant to environmental factors such as elevated temperature or humidity. Moreover, PLA/CLS composites show a higher flow rate and higher surface hydrophobicity in comparison with unmodified PLA.

## 1. Introduction

Lake sediments contain, in various proportions, carbonate and non-carbonate mineral ingredients and an organic ingredient. The non-carbonate mineral component includes terrigenous elements and those of biogenic origin. The organic component contains elements of plant and animal origin and products of their metabolism and decomposition [1]. Mineral matter in the lake is a product of chemical denudation delivered with ground and surface waters and a product of mechanical denudation, mainly derived from erosion processes [2]. The predominant sediment in Lake Swarzędzkie is calcareous gyttja, with varying carbonate contents. This sediment fills most of the lake basin. In the lake sediments, the predominant component is gyttja, with a calcium carbonate content of more than 80%, called lacustrine chalk, lying in the southern part of the water body under a 2- to 3-m turf layer. The colour and consistency of the gyttja and lake chalk changes with the depth. The deeper the sediment, the darker their colour and the lower the water content.

Down to a 12 m depth, two mineral phases can be observed. One is SiO_2_ with a quartz structure and the other is CaCO_3_. As the sampling depth increases, mineral phases of kaolinite, muscovite, mullite, and anhydrite also appear [3]. In lakes, the surface layer of sediment is usually hydrated at about 80–95%. Hydration decreases with depth due to sediment compaction and reaches 65% at a depth of 5 m [4].

Bottom sediments, just like other materials of natural origin, can be used successfully as composite fillers. Natural fillers include, among others, microcrystalline cellulose, cellulose fibres, wood [5], and amorphous diatomaceous earth [6,7]. This not only improves a range of physicochemical properties but also increases the biodegradability of the composite. In our previous studies, the composite matrix was PLA.

PLA (polylactide acid, polylactide) is one of the main materials on the market of biodegradable polymers [8,9,10]. PLA is a linear aliphatic polyester that is formed by the polymerisation of lactic acid monomers or oligomers (especially the dimers-lactides). Monomers are obtained by the fermentation of starch, e.g., from maize or sugar beet [11,12].

Besides that, enzymatic polymerisation in mild conditions is also used as an environmentally friendly alternative [13]. Another method of polylactide manufacturing is microbiological fermentation, which is advantageous on account of the lack of extraction and purification stages necessary in other processes. Microbiological polylactide synthesis is a one-step process allowing for the control of the structure of the polymer by regulating the ratio of lactide monomers [14,15]. Polylactide is often used as packaging material, but in addition to packaging applications, biopolymers can be utilised in a variety of industries, such as agriculture, automotives, construction, and electronic equipment. Polylactide, like other biopolymers, has its weaknesses when it comes to certain properties, e.g., low impact strength, low gas barrier characteristics, low thermal resistance, and low rate of crystallisation [16,17,18]. Methods to modify polymers are therefore sought by introducing, e.g., (nano) fillers, plasticisers, and flame retardants into the polymer matrix [19,20,21,22,23,24,25].

Lake sediments are a potentially interesting replacement for mineral fillers in polymer and biopolymer composites. Their use may be dictated by the possibility of the reclamation of water reservoirs undergoing adverse anthropogenic changes. The conducted research is to indicate the potential direction of application as well as differences or advantages over conventional fillers.

In this work, we used raw bottom sediments collected directly from Lake Swadzędzkie as a polylactide modifier. Field studies were carried out in September 2017 within Lake Swarzędzkie. Lake Swarzędzkie is situated in western Poland, in the Wielkopolsko-Kujawski Lake District. It is a shallow, eutrophicated water body with an area of approx. 0.94 km^2^, with the river Cybina flowing through it.

The tests were carried out to investigate the effects of the natural filler on the mechanical and processing properties of the produced polylactide/carbonate lake sediment (PLA/CLS) composites. To determine the characteristics of the sediment, phase analysis (XRD), analyses of particle size distribution by DLS (Dynamic Light Scaterring), thermogravimetric analysis (TGA), and infrared spectrophotometry (FT-IR) were carried out. In addition, for the PLA/sed A-B composites, tests were performed on the melt flow rate (MFR), strength (tensile strength, flexural strength, impact strength), and hydrophobic properties of the surface. Strength tests were conducted for the composite samples before and after weathering tests. The filler/PLA matrix dispersion was assessed via optical and electron microscopy.

## 2. Materials and Methods

### 2.1. Materials

Polylactide (PLA Ingeo™ Biopolymer 2003D, NatureWorks) (Minnetonka, Minneapolis, MN, USA) was obtained as a matrix polymer. The sediments were collected from Lake Swarzędzkie in September 2017 at the geographical coordinates 52°24′50.5″ N; 17°04′14.8″ E.

### 2.2. Samples Preparation

#### 2.2.1. Preparation of Sediments

To identify the thickness and the spatial diversity of the sediments in Lake Swarzędzkie, eight boreholes were made in the bottom of the water body up to a depth of 15 m under the water surface, with a total length of 65.5 linear metres. The sediments were collected in 1-metre and 0.5-metre cores with an Ins-torf drill and a specialised boat for drilling in the bottom of the lake. Based on the distribution of thickness and sediment types, a 14-metre core with the geographical coordinates 52°24′50.5″ N; 17°04′14.8″ E, representing the thicker area of the water body, and consisting of turf and carbonate sediment was selected for further examination. The core was divided into 16 parts, depending on the depth of the sediment collected, every 1 m on average.

Two sediment fractions were selected for further investigation, namely, from the depth of 3–8 m (sediment A) and from the depth of 8–12 m (sediment B). The combined sediments were ground in the ball mill for 72 h and then fractionated using a vibrating sifter with sieves with different mesh sizes (>40 µm and <40 µm). For further testing, sieved sediment fractions with a particle size below 40 µm were used.

#### 2.2.2. Preparation of Masterbatches

The process of homogenising polylactide (PLA) with fractionated sediments was carried out with the laboratory two-roll mill ZAMAK MERCATOR WG 150/280 (ZAMAK, Skawina, Poland). A total of 1000 g of PLA was plasticised at 215 °C over the duration of 15 min while adding portions of sediments until a filler concentration of 50% by weight was reached. Then, the batches were ground with the mill WANNER C17.26 SV (WANNER, Wertheim, Germany) and dried for 24 h at 60 °C.

#### 2.2.3. Preparation of Final Samples

Ready master batches were diluted with neat PLA directly in the injection machine Engel e-victory 170/80 (Engel, Schwertberg, Austria), until concentrations of 2.5%, 5%, 10%, and 15% were reached. Table 1 shows the injection moulding parameters. Standardised test specimens according to PN-EN ISO 20753:2019-01 were obtained for mechanical tests. The final concentrations in the batches are shown in Table 2.

### 2.3. Characterisation Methods

The phase identification and the relationship between the sediment fraction depth and its composition were determined using an X-ray diffraction (XRD) powder diffractometer Bruker AXS D8 Advance (Bruker, Karlsruhe, Germany) using CuKα lamp radiation and an Ni filter. X-ray diffractograms were recorded in the angular range of 20–80° [2Θ].

Thermogravimetry (TG) was performed using the gravimetric analyser NETZSCH 209 F1 Libra (NETZSCH, Selb, Germany). Samples of 5.0 ± 0.2 mg were cut from each granulate and placed in Al_2_O_3_ crucibles. Measurements were done under air in a 30–1000 °C temperature range and at a 10 °C/min temperature rise.

Differential scanning calorimetry (DSC) was performed using a NETZSCH 204 F1 Phoenix (NETZSCH, Selb, Germany). Calorimeter samples of 3 ± 0.2 mg were cut and placed in an aluminium crucible with a punctured lid. The measurements were performed under nitrogen in the temperature range of 20–200 °C and at a 10 °C/min heating rate. The measurements were carried out in two cycles.

Fourier Transform-Infrared (FT-IR) spectra were recorded on a Nicolet iS50 Fourier transform spectrophotometer (Thermo Fisher Scientific, manufacturer Madison, WI, USA) equipped with an ATR unit (5000–80 cm^−1^).

The particle size distribution of the filler used to prepare the composites was measured with a Mastersizer 3000 (Malvern Instruments Ltd., Malvern, UK), working under the Dynamic Light Scattering (DLS) principle.

The measurements were made for the samples in water suspension (Hydro EV attachment). The parameters of the measurements for the wet method samples were: stirrer speed-2330 rpm, ultrasound power-70%.

Contact angle analyses were performed using the sessile drop technique (5 μL) at room temperature and atmospheric pressure with a Krüss DSA100 goniometer (Krüss Optronic GmbH, Hamburg, Germany).

Images of composite fractures were taken with the digital light microscope KEYENCE VHX-7000 (KEYENCE INTERNATIONAL NV/SA, Osaka, Japan) with a VH-Z100R wide angle zoom lens (KEYENCE INTERNATIONAL NV/SA, Osaka, Japan). at 1000× magnification. The images were taken using the function of depth composition and 3D image creation. Standard coaxial lighting was used.

Surface imaging was performed with SEM electron microscopy (Quanta 250 FEG, FEI Thermo Fisher Scientific, Waltham, MA, USA).

The ageing tests were performed in the ESPEC ARS-0220 (ESPEC, Pyeongtaek, Republic of Korea) weathering chamber with ten alternate cooling and heating cycles (−10 °C to +50 °C) and at a humidity of 85% (humidity control is applicable only at temperatures above approx. 10 °C).

For flexural and tensile strength tests, the materials obtained were printed into type 1B dumbbell specimens in accordance with the EN ISO 527:2012 and EN ISO 178:2006 standards. The tests of the obtained specimens were performed on a universal testing machine Instron 5969 (Instron, Norwood, MA, USA) with a maximum load force of 50 kN. The traverse speed for both flexural and tensile strength measurements was set at 2 mm/min.

The Charpy impact test (unnotched samples) was performed on an Instron Ceast 9050 impact tester (Instron, Norwood, MA, USA) in accordance with the ISO 179-1:2010 standard.

The measurement of the mass melt flow rate (MFR) was carried out according to the PN-EN ISO 1133 standard with the Instron CEAST MF20 (Instron, Norwood, MA, USA) extrusion plastometer at 210 °C and using a 2.16 kg load.

The dynamic viscosity coefficient was determined by a capillary rheometer Instron Ceast SR 10 (Instron, Norwood, MA, USA), according to the ISO 11443:2005 standard, using a capillary tube of 5 mm in length and 1 mm in diameter and a shearing speed range of 1−100 (s^−1^) at 190 °C.

## 3. Results and Discussion

### 3.1. XRD Analysis of the Sediment

Among the sediments found in Polish lakes, there is a sediment group formed biochemically or chemically. These are mainly rocks made of calcium carbonate, the content of which is between a few to more than 95% [26]. To identify the crystalline phase of the A and B sediments, X-ray diffraction was used, and the results are shown in Figure 1. After treating the samples at 500 °C in a muffle furnace for 30 min in an air atmosphere, the diffractograms of the sediments A and B do not show any differences; therefore, only the results obtained for sediment B were presented. The main phase of the sediment is CaCO_3_ (calcite), with reflections at 2θ values corresponding to it: 23.0; 29.5; 31.5; 36; 39.5; 43.2; 47.15; 47.5; 48.5; 48.9; 56.8; 57.4; 58.1; 60.7; 61.0; 61.4; 63.1; 64.7; 65.7; 68.2; 70.3; 72.9; 73.7; 76.4. The other identified phase in the sediment is SiO_2_ with a quartz structure. This is indicated by the presence of reflections at the following 2θ values: 26.6; 31.7; 58.0; 61.0; 70.4; 71.6. The X-ray analysis is confirmed by infrared spectroscopy tests, which helped identify the presence of the bands specific to CaCO_3_.

### 3.2. FT-IR Analysis of the Sediments

Figure 2 shows the FTIR spectra of sediments A and sediment B.

The bands specific to carbonates include four areas: (ν1) symmetric stretching at a wavelength of 1080 cm^−1^, (ν2) out of-plane bending at a wavelength of 870 cm^−1^, (ν3) doubly degenerate planar asymmetric stretching at 1400 cm^−1^, and (ν4) doubly degenerate planar bending at a wavelength of 700 cm^−1^ [27]–Table 3. The FTIR analysis of the sediment A and sediment B samples showed the presence of all the bands characteristic of calcium carbonate at the following wavelengths: 1411, 1088 and 1029, 864, and 705 cm^−1^.

### 3.3. The Measurement of Particle Size Distribution by DLS (Dynamic Light Scaterring)

The ground and sieved sediments were tested for particle diameter distribution. The measurements carried out showed differences between the two tested sediments (Figure 3). For small particles (0.2–1 µm), the curves are similar for both sedA and sedB. Above 1 µm, for sedimentB, we have a slightly larger proportion of particles of 2 µm and of 8–9 µm. No particles larger than 25 µm are present. On the other hand, for sedimentA, the highest percentage is observed for particles of 6 µm (8 µm for sedB), and particles between 25 µm and 100 µm are also present, which may indicate a tendency toward the agglomeration of sedA particles. Variations in the distribution of the filler particles used in the modification of polylactide may affect both the mechanical strength of the composites and their impact strength. This may be due to differences in the dispersion of the filler in the polymer matrix, which affect the formation of agglomerates. Fillers with small particle sizes are well suited as polymer modifiers due to the potentially high dispersion ability in the matrix.

### 3.4. Thermal Behaviour Properties of the Sediment-TGA

The results of the thermal analysis of the bottom sediments are shown in Figure 4. The curves are almost identical for both sediment fractions. The presence of three peaks at ~80 °C, 301.7/304.2 °C, and 731.2/736.0 °C, respectively, indicates a three-stage process of sediment decomposition. The first stage is related to the loss of physically adsorbed water. At the second stage, the loss in mass is 5.58% and 5.64% for sedA and sedB, respectively. This process corresponds to the combustion of organic compounds. On the other hand, at approx. 680–690 °C, the thermal decomposition of CaCO_3_ occurs (Table 4) [28,29]. The total loss of mass for both filler fractions is approx. 50%. The results of the thermal analysis, in addition to the X-ray analysis and FTIR, confirm the same composition of both sediment fractions, regardless of the depth of their collection.

### 3.5. Rheology

#### 3.5.1. Melt flow Rate (MFR)

The melt flow rate was determined for both neat PLA as a reference sample and for composites on the PLA matrix modified with sediments (Figure 5). The measurements showed that the addition of sediments, regardless of the concentration, affects the values of the melt flow rate and, as a result, leads to changes in the processing parameters of materials. The biggest improvement in the melt flow was achieved for batches containing sediment A (3–8 m). As the concentration of the additive increased, the MFR grew to 16.8 g/10 min for a batch containing 10% of the modifier. This concentration of the additive is optimal due to a decrease in the MFR value with a further increase in concentration. Batches modified with sediment B behave differently—as the concentration increases, the melt flow rate decreases, while at a concentration of 15%, it increases again to reach 13.1 g/10 min.

Taking into account the composition of the two sediment fractions, the factor affecting the MFR is the particle size of the fillers used and their tendency to agglomerate. SedA, from fractions located closer to the surface, has a higher tendency to agglomerate and a higher content of organic matter (Figure 4), which increases the MFR, although this happens up to 10% of the filler content. SedB, from deeper fractions, which is transformed and contains a lower proportion of organic compounds, shows an increased MFR at the concentration of 15%.

The MFR measurements show that sediment fillers have a positive effect on the melt flow of the obtained compositions when compared to the neat PLA (Figure 5), which was also previously observed for PLA/diatomaceous earth composites [6,7]. Sediment A was more effective for that matter, showing an increasing effect up to 10% loading, while sediment B increased the flow rate significantly only at 15%. The observed differences may be linked to the quantitative differences between the fillers, as sediment B consists of a larger number of particles of a higher average size, as observed from DLS measurements. The increase in MFR values may be caused by the slight decrease in the average molecular mass of PLA chains due to polymer degradation during melt processing.

#### 3.5.2. Viscosity

The highest viscosity at a shearing rate from 0 to 1000 s^−1^ was recorded for the reference PLA, which indicates that the addition of sediments of any type will reduce the viscosity of composites (Figure 6). With the increase in the shearing rate, a steep decrease in the PLA viscosity curve is observed, while modified composites, especially those with sedA, have a relatively constant viscosity in the range from 200 to 400 Pa·s. This is similar for sedB deposits, however; the viscosity range (200–600 Pa·s) and its decrease are greater. For sedA, as the concentration of the filler increases, the viscosity decreases gradually, while for sedB, the curves are slightly different—composites containing 2.5 and 15% of the filler have the lowest values, while those with a medium concentration have higher and almost identical values. For this filler, the effect of thinning by shearing is more pronounced [30]. It is clear from the above that even a small addition (2.5%) of sediment as a filler has a significant effect on the viscosity of composites. Sediments containing untransformed organic compounds (sedB) reduce the viscosity of composites.

### 3.6. DSC Analysis

A DSC analysis is performed to investigate the thermal behavior of the PLA and PLA/sediment composites. Figure 7 shows the DSC of the PLA and PLA/sediment composites during the first and second heating stages. Each DSC curve shows three characteristic events: (a) a glass transition (T_g_) (60–70 °C), (b) cold crystallization (T_cc_) (110–130 °C), and (c) melting (T_m_) (145–155 °C) (Figure 7 and Figure 8).

The addition of sediments to the polymer did not significantly affect the glass transition temperature or the melting point for both the samples with the sediments 3–8 and 8–12. In the PLA/sediment composite materials, a large glass transition peak is noticeable in the first heating cycle, which is related to the low crystallinity of the polymer (high proportion of the amorphous phase). The introduction of the filler did not affect the glass transition effect significantly, showing little matrix–filler interaction. On the other hand, significant changes in the modified materials with sediment fillers can be noticed at the cold crystallization event (T_cc_) in relation to the neat PLA sample. The curve for the reference sample has a broad T_cc_ signal due to the fact that PLA is a semi-crystalline polymer with a slow crystallization rate. The applied fillers resulted in a significant narrowing of the cold crystallization signal, as well as a decrease in the T_cc_ with an increase in the sediment content. These results indicate that the deposits in the polylactide matrix exhibit the nucleating properties. The sediment A performs better in this function, as evident by more narrow and prominent T_cc_ and T_m_ signals. During the first heating cycle, small right-side shouldering of the melting signal is visible, which is either a result of the melting of two types of crystallites or the effect of metastable crystal perfecting and remelting. During the second cycle, there is only one sharp signal, confirming the formation of only one type of crystallites.

### 3.7. Mechanical Properties

The introduction of both types of powder fillers to the polylactide matrix increases the Young’s modulus of composites and, at the same time, decreases their strength and elongation at break compared to PLA (Figure 9 and Figure 10). For example, the Young’s modulus containing 10% of sediments per weight increases by approx. 523 and by approx. 357 MPa for fillers A and B (Figure 9). For the same materials, the strength and elongation at break decrease by approx. 5.0 MPa and approx. 0.66% for sediment A and by 6.5 MPa and approx. 0.69% for sediment B.

The improved values of the Young’s modulus compared to the non-modified PLA are due to the introduction of rigid fragments, i.e., powder fillers, into the polymer matrix. At the same time, these fragments interfere with the continuity of the polymer matrix and reduce the elongation at break. Discontinuities in the material structure of composites caused by the presence of powder filler, referred to as microcarbs, are places of stress concentration. In these areas, material decohesion begins in the form of developing micro-cracks. An additional stress concentration, which causes material cracking, may occur in the interphase polymer-filler area. The decrease in the elongation at break of the tested composites may be due to the poor adhesion of the applied fillers to the polymer matrix, the poor dispersion of their particles in the non-polar polyolefin matrix, and the ability to form agglomerates that have low adhesion to the matrix [31,32]. The inorganic powder filler may also be unable to transmit the stresses carried by the polymer matrix, which also results in the observed decrease in elongation.

The increasing content of sediments gave a gradual increase in the Young’s modulus of polylactide-based composites while reducing their tensile strength at the same time (Figure 10). Neat PLA has a Young’s modulus of approx. 3.3 GPa. For composites containing 15% of fillers by weight, the Et increases to reach approx. 3.9 and 3.7 GPa, respectively, for sediments A and B. According to Nyambo C. et al. [33] the observed increase in the Young’s modulus in composites compared to the polymer matrix, growing with the increase in the filler content in the matrix, may result from an increase in the crystallinity of composite samples. The inverse relationship was recorded for tensile strength. The increase in the content of both types of sediments used causes its gradual decrease, but only to a small extent. The tensile strength of neat PLA is approx. 66.4 MPa. On the other hand, with 15% of the filler by weight, this value is reduced to 59.9 and 58.5 MPa, respectively, for sediments A and B. According to Fu [34], this tendency is characteristic for polymer composites containing non-modified fillers with particle sizes above 1 µm average diameter. It may also be the result of the weak hydrophilic bonding of the powder filler with the hydrophobic polymer matrix and the poor dispersion of the powder filler in the polymer structure [32,35]. Agglomerates result from the tendency of micro-particles to be attracted to one another by electrostatic forces and van der Waals forces. When a tensile load is applied to low filler composites, there is an efficient stress transfer between single particles and the matrix. However, the stress transfer becomes less efficient when particles are agglomerated [36]. The increasing sediment content in the polylactide matrix is accompanied by a gradual decrease in elongation at break—from 2.89% (neatPLA) to 2.20% for the composite 15sedA and 2.08% for 15sedB. The polymer matrix plasticity decreases as a result of the reduced mobility of polymer chains caused by the introduction of powder fillers [32]. Similar relationships were also obtained in a number of works dedicated to PLA composites containing other powder fillers: eggshell powder [36], chestnut shell [37], coconut shell [32], olive pit powder [38], wood flour, and talc [35].

As a function of the depth of extraction of the fillers used, it was observed that slightly higher values of the strength properties are characteristic of composites containing sediments collected at the depth of 3–8 m. These differences are small and more pronounced for materials containing 10 and 15% of the filler by weight, which is probably due to variations in the particle size distribution of both fillers.

As for impact strength, no significant changes were observed as a result of adding sediments to the polylactide matrix (Figure 11). The resulting impact strength values for pure PLA and its composites focus around comparable values and are within the mutual confidence intervals of the standard deviation. According to [38], neat PLA is a brittle material, and the addition of powder filler to its structure intensifies this property.

The flexural strength results confirm all the above-described dependencies (Figure 12). Higher flexural modulus values were obtained for composites compared to pure PLA. This modulus also increases with the sediment content in the polylactide matrix, from approx. 3.5 GPa (PLA) to 4.3 GPa for 15 sedA and 4.4 GPa for 15 sedB.

Similarly, as the filler content in the polymer structure increases, the elongation of the composites subject to the bending test decreases (Figure 12). For pure PLA and its composites containing 2.5 and 5% of A and B fillers by weight, the test was carried out until the specific bend deflection was reached. For the filler content of 10 and 15% per weight, the samples were destroyed before this parameter was reached, which suggests a low adhesion of the filler to the polymer matrix, a poor sediment dispersion in the PLA structure, and an affinity for agglomerating.

The results obtained in the mechanical properties tests are typical for composites with mineral (chalk) fillers. The A and B sediments are suitable for use as fillers for thermoplastic polymers due to the absence of a significant deterioration of mechanical properties.

### 3.8. Mechanical Tests after Conditioning in the Weathering Chamber

The strength tests for PLA/CLS composites after conditioning in the weathering chamber are shown in Figure 13, Figure 14, Figure 15 and Figure 16.

The conditioning of PLA/sedA-B composites in the weathering chamber, where they were exposed to temperatures from –10 °C to 50 °C, resulted in changes in their mechanical characteristics. Pure polylactide’s structure is weakened, resulting in a decreased impact strength from ~19 to ~14 kJ/m^2^ and increased brittleness (the elongation at break decreased from ~2.9 to ~2.7%). The Young’s modulus also decreased slightly (from 3.3 GPa to 3 GPa), which may be associated with the reduced crystallinity of the polylactide. Decreased parameters were also observed for flexural strength. However, all the tested composites after conditioning in the weathering chamber have an impact strength and tensile values similar or even higher. The highest increase in these parameters was observed for composites containing sedA. The decreased parameters in the pure polylactide and the increased parameters in the sedA-B composites suggest the degradation of PLA due to the increased temperature and humidity, while modification with sediments has a positive effect on the strengthening of the composite structure. The impact strength increased to approx 26 kJ/m^2^ for the 10% content of the sedA sediment, while a 50% increase in flexural strength was observed for the 15 sedA composite compared to the unmodified PLA.

### 3.9. Composite Surface Characterisation

#### 3.9.1. Composite Wettability Tests

The results of the contact angle measurements for the PLA/CLS composites are shown in Figure 17. The application of bottom sediments as fillers changed the hydrophobic properties of the polylactide. The highest contact angle values were obtained for the batches modified with the sediment collected from fractions closer to the surface (3–8 m)—sediment A. The contact angle for the 15% filler content is 98.3°, while that for the additive content of 2.5% is 89°. The modification of the polylactide with sediment B showed slight differences in hydrophobic properties between samples with 2.5% and 15% of CLS content (87.5° and 88.4°, respectively), and these are the highest results in the series. Therefore, it can be concluded that, regardless of the amount of filler, it is possible to obtain surfaces less prone to wetting.

The contact angle tests after the samples were placed in the weathering chamber showed (Figure 13) that the values are lower for pure polylactide than they are for the samples conditioned at room temperature (the contact angle is lower by 3°). On the other hand, for sediment-modified batches, higher values are obtained for lower concentrations of sediment A (3–8 m). As the concentration increases, the hydrophobic properties of the surface decrease, which is caused by the influence of increased temperature and humidity on the properties of the sediment itself. For batches modified by the sediment collected from deeper layers (sediment B), increased hydrophobic properties are obtained for batches containing 5 and 10% of the additive; however, this increase does not give a fully hydrophobic surface, as is the case with the sedA samples.

Figure 18 presents example photos of droplets applied to the surface of the tested samples. In addition to the changed shape of the droplet, a slight change in surface roughness is observed. The presence of sediment B in the composite clearly increases surface roughness and creates hydrophobic centres, which significantly increases the contact angle.

#### 3.9.2. Microscopic Observations

The EDS mapping of composite fractures showed that the samples modified with sedA as well as with sedB contain concentrations of calcium compounds unevenly distributed in the composite (Figure 19). Their content changes as the filler concentration increases, and there are more of them for 5sedB than for 5sedA. The other elements present in the samples are carbon, oxygen, aluminium, silicon, and traces of iron. In batches with lower concentrations of calcium, carbon is predominant over calcium, but as the filler content increases, this tendency is reversed.

On the SEM images of composite fractures before (Figure 20A–D) and after the weathering chamber (Figure 20A’–D’), differences in the structure of the materials analysed were observed. They are best visible for the composites modified with sedA (Figure 20A–2.5sedA, Figure 20B–15sedA). Conditioning in the weathering chamber causes changes in the structure of the fracture surface visible as cracks in the materials. This effect is noticeable as the filler concentration increases. The surface of the composite has places with a milky colour, which are associated with the degradation of polylactide occurring at high temperatures and in the presence of moisture. At the same time, a much better dispersion of the filler in the matrix is observed, especially for the 15% sedA content. For the composites modified with sedB, after conditioning in the weathering chamber, no changes in the surface structure were observed, apart from the milk-coloured areas. This suggests that the addition of sedB stabilises the material.

The SEM images of composite fractures (Figure 20) showed the presence of filler particles of a size corresponding to the sizes observed in the DLS measurement (Figure 3). For booth composite types, both particles of 0.2–10 μm and of approx. 10 μm are present. No filler agglomerates were observed, which indicates their good dispersion in the polymer (Figure 21).

## 4. Conclusions

The bottom sediments collected from Lake Swadzędzkie contain mainly calcium carbonate in their composition, and, as a result, they can be used as composite fillers. In the study, sediment and PLA composites were obtained. Two fractions from 3–8 m and 8–12 m were used for this purpose, in quantities of 2.5, 5, 10, and 15% by weight. PLA/CLS composite strength tests showed varied effects of the fractions. The increased proportion of the filler causes the Young’s modulus to increase, but the elongation and tensile strength decreases with the growing filler proportion. Higher flexural modulus values were also obtained for the composites compared to the pure PLA. This modulus also increases with the increase in sediment content in the polylactide matrix. Similar effects were observed for samples conditioned in the weathering chamber, especially when sediment A was the filler.

## Figures and Tables

**Figure 1 materials-15-06106-f001:**
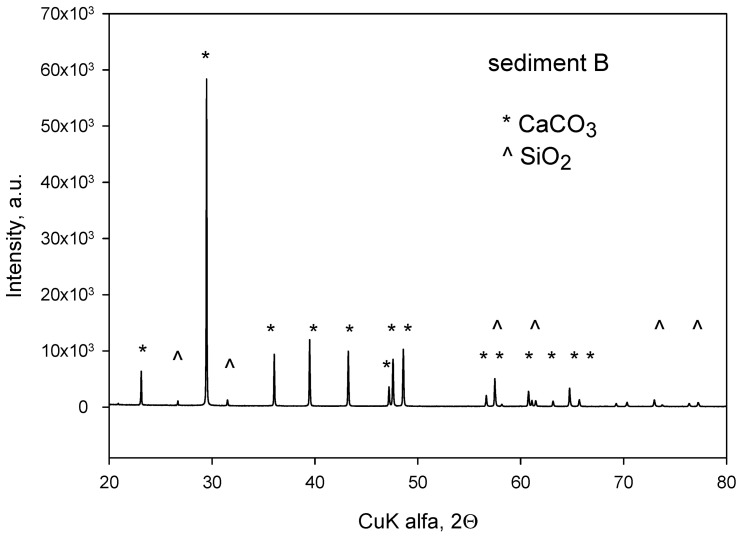
XRD diffractogram of sediment B (after thermal treatment at 500 °C).

**Figure 2 materials-15-06106-f002:**
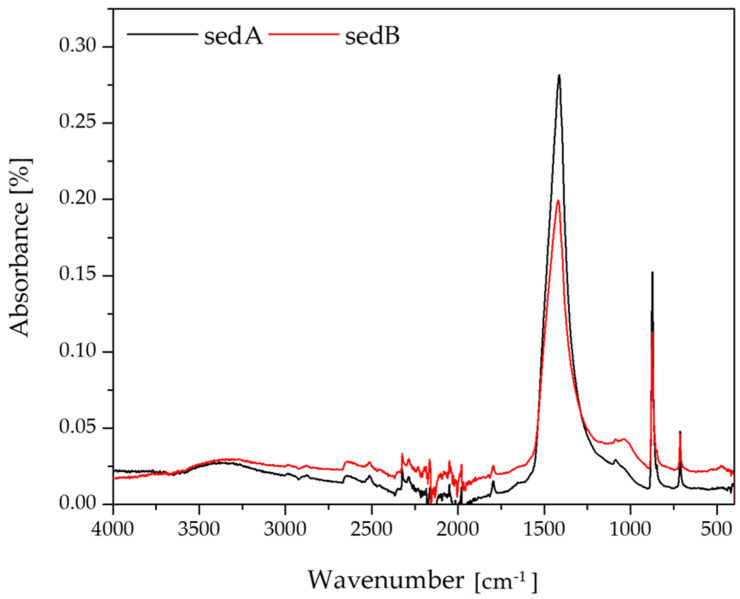
FT-IR spectra of sediments.

**Figure 3 materials-15-06106-f003:**
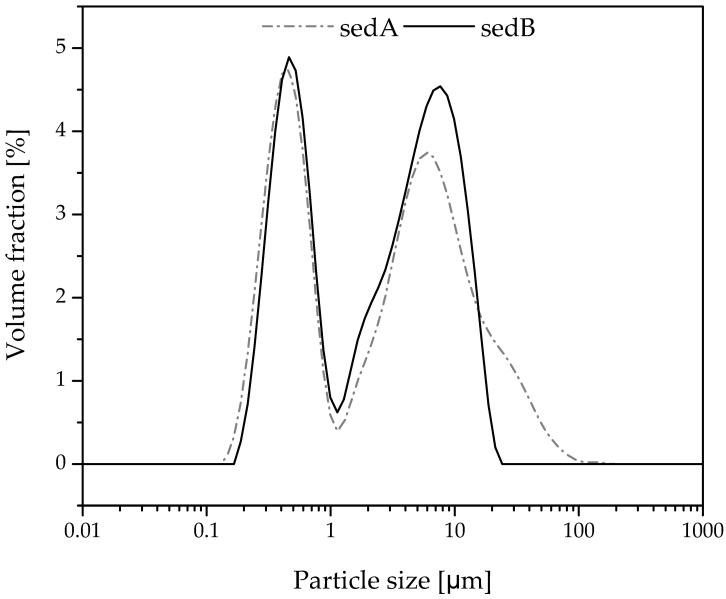
Distribution of particle sizes of fractionated sediments.

**Figure 4 materials-15-06106-f004:**
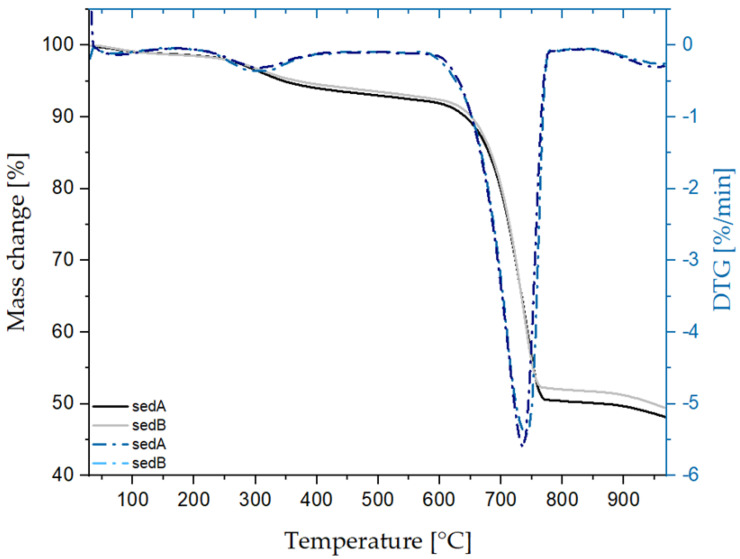
Thermogravimetric and Derivatographic Curves of the Sediments.

**Figure 5 materials-15-06106-f005:**
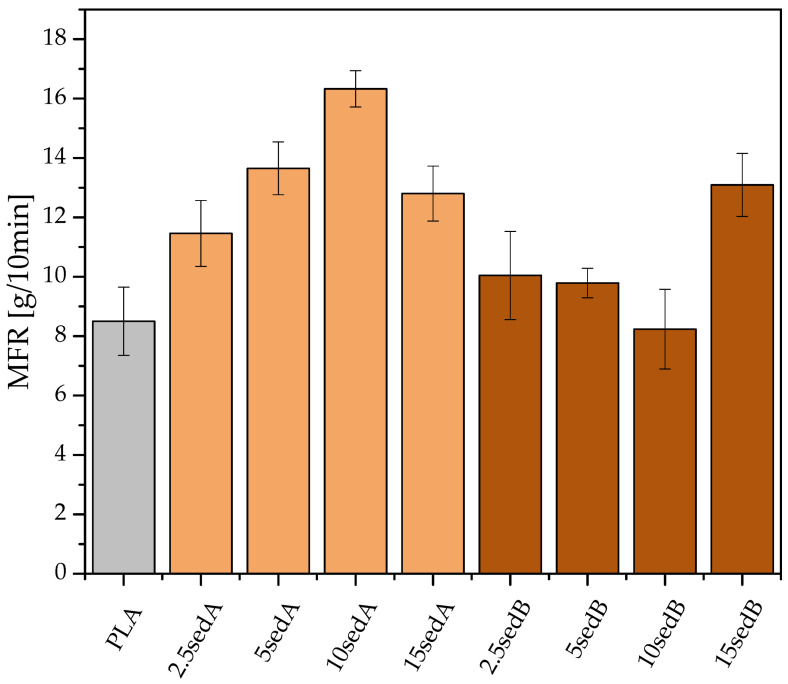
Results of the Mass Melt Flow Ratio Measurements.

**Figure 6 materials-15-06106-f006:**
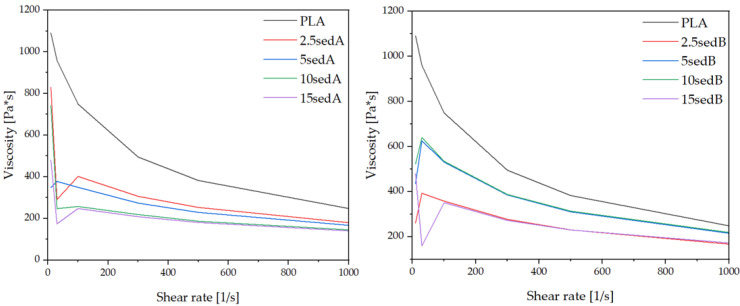
Viscosity of PLA/CLS composites after injection moulding.

**Figure 7 materials-15-06106-f007:**
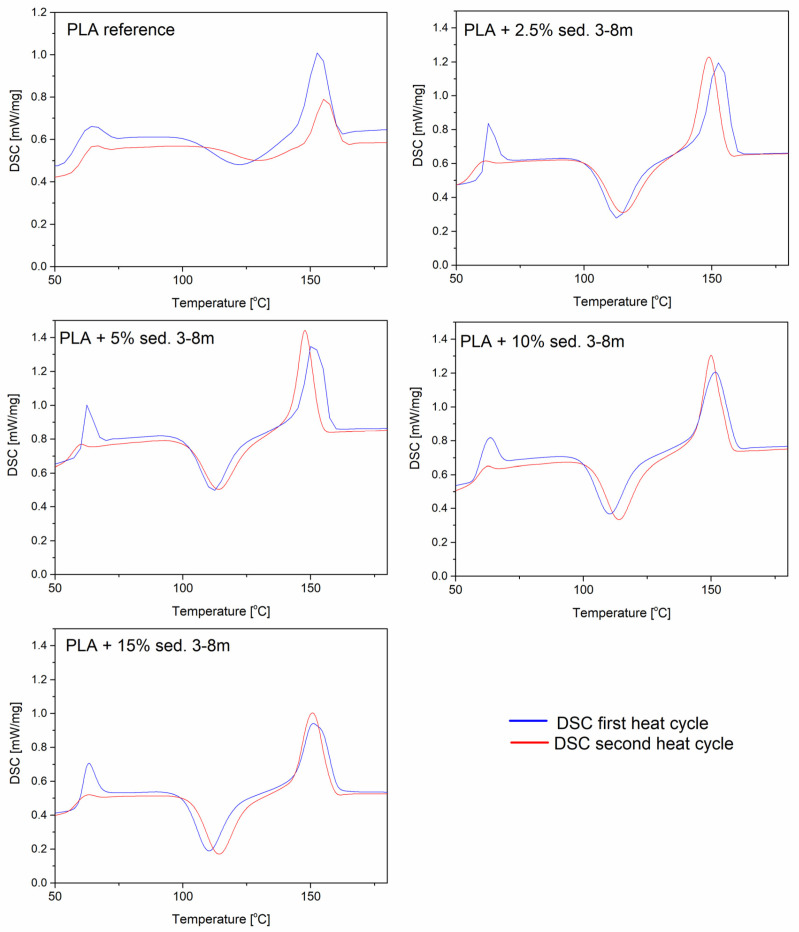
DSC plots of PLA/sedA.

**Figure 8 materials-15-06106-f008:**
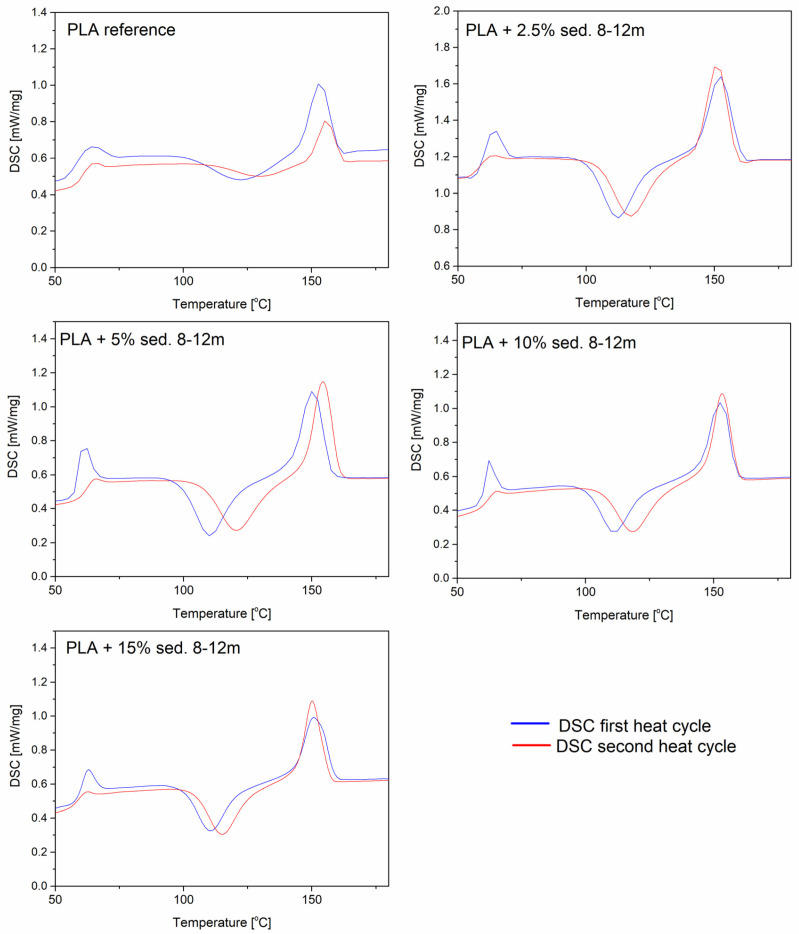
DSC plots of PLA/sedB.

**Figure 9 materials-15-06106-f009:**
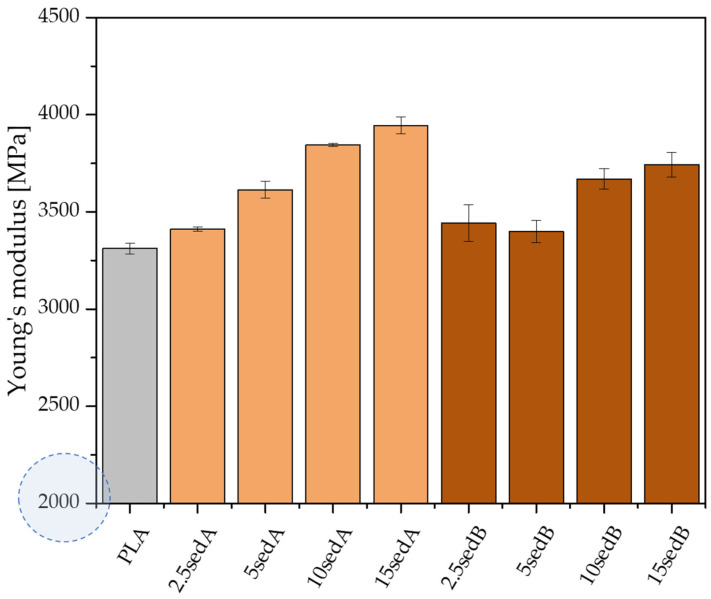
The Young’s modulus for PLA/sedA-B composites.

**Figure 10 materials-15-06106-f010:**
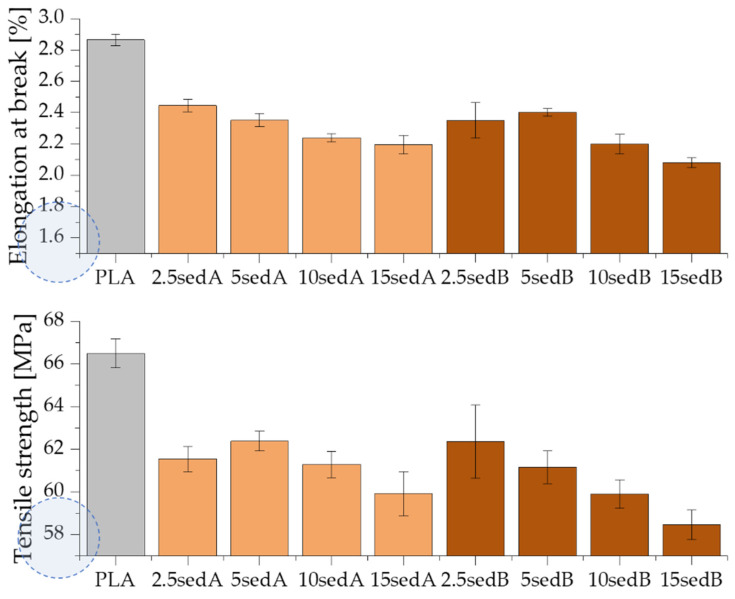
Tensile strength and elongation at break for PLA/sedA-B composites.

**Figure 11 materials-15-06106-f011:**
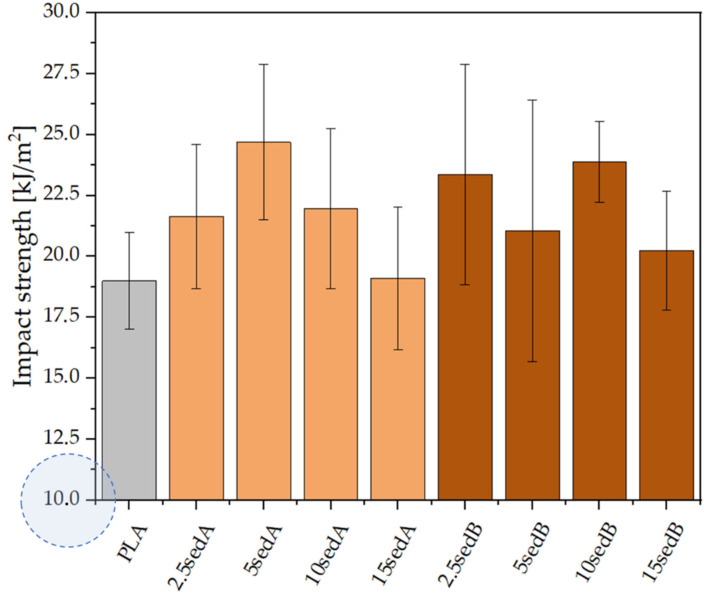
Impact strength of the PLA/sedA-B composites.

**Figure 12 materials-15-06106-f012:**
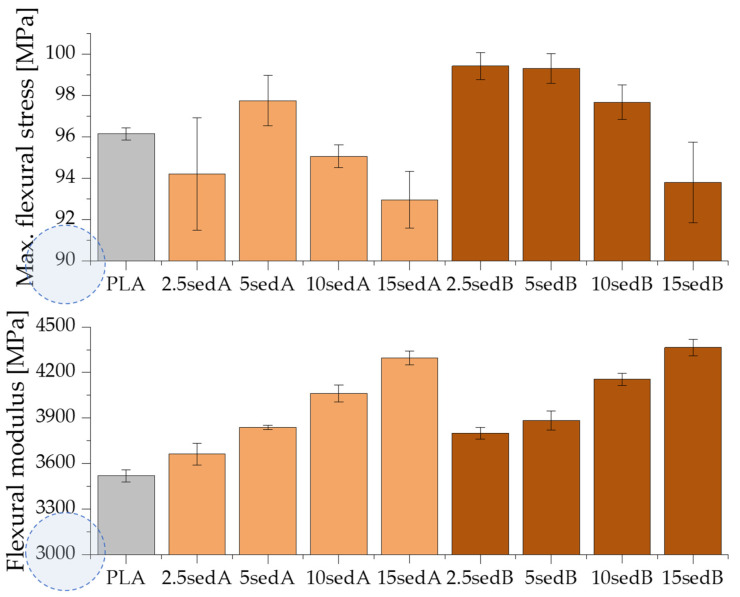
Flexural strength of PLA/sedA-B composites.

**Figure 13 materials-15-06106-f013:**
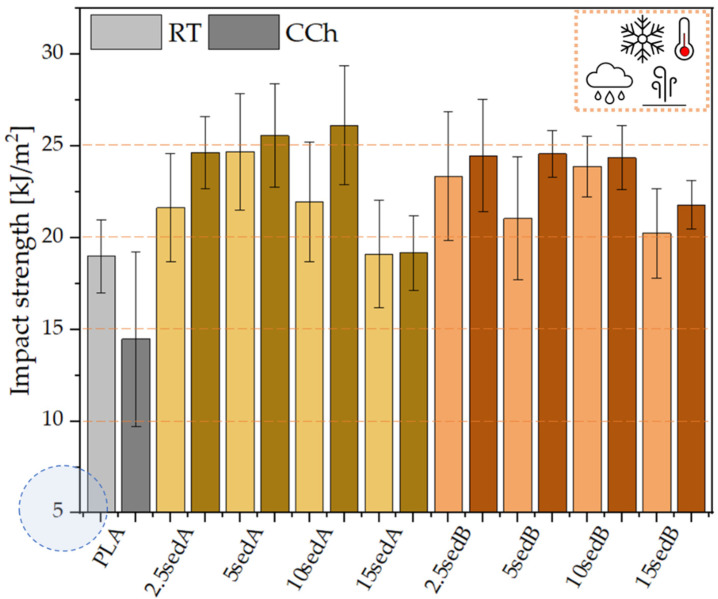
Impact strength for composites after conditioning in the weathering chamber (WCh).

**Figure 14 materials-15-06106-f014:**
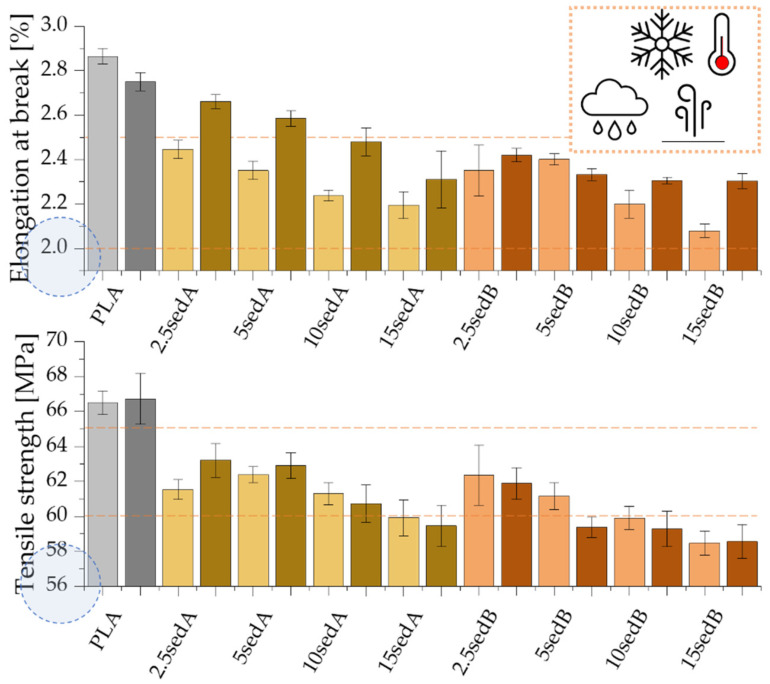
Tensile strength and elongation at break for the composites PLA/sedA-B after conditioning in the weathering chamber (WCh).

**Figure 15 materials-15-06106-f015:**
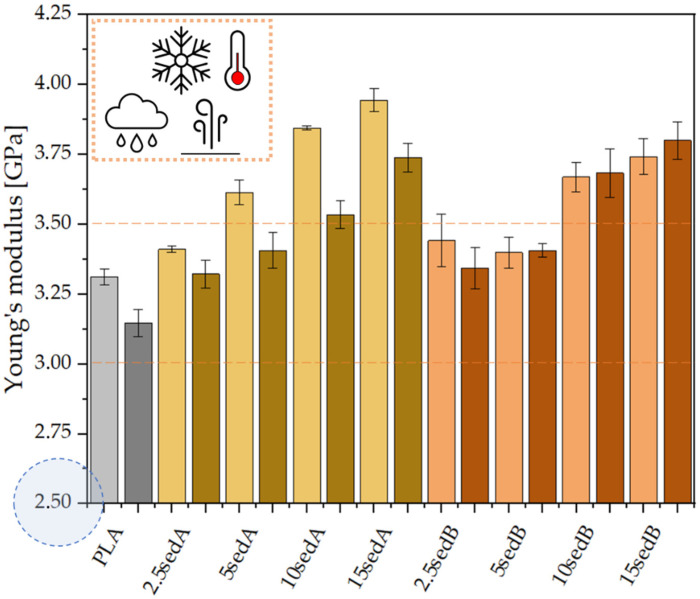
Young’s modulus for the composites PLA/sedA-B after conditioning in the weathering chamber (WCh).

**Figure 16 materials-15-06106-f016:**
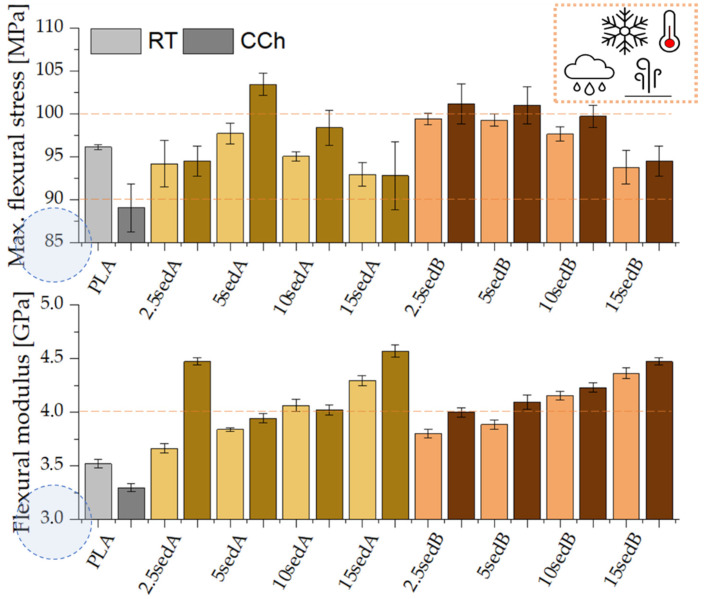
Flexural modulus and max. flexural stress for the composites PLA/sedA-B after conditioning in the weathering chamber (WCh).

**Figure 17 materials-15-06106-f017:**
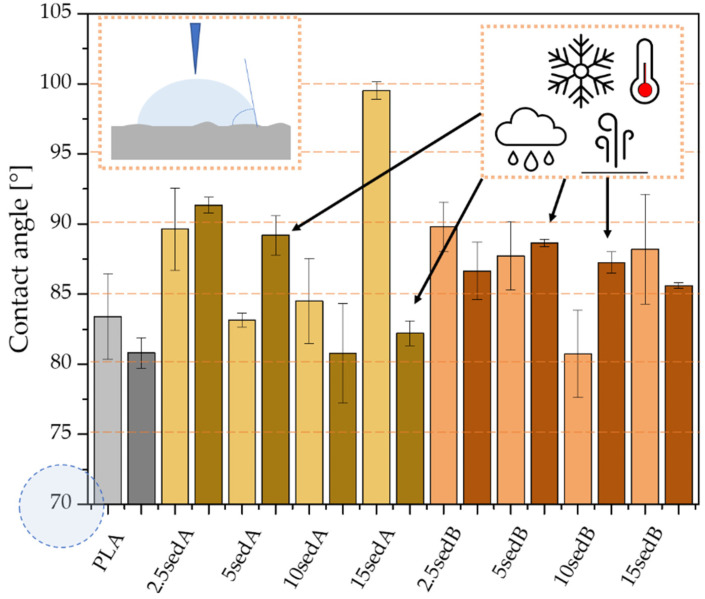
Contact angle for the composites PLA/sedA-B after conditioning in the weathering chamber (WCh).

**Figure 18 materials-15-06106-f018:**
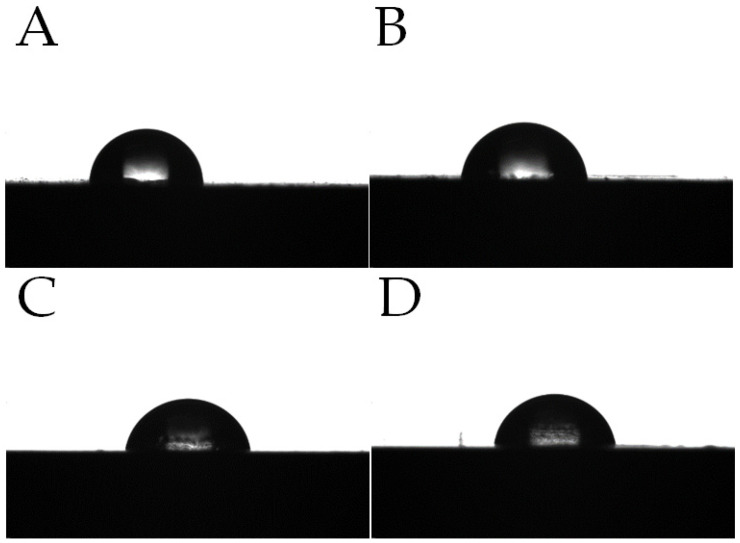
Water droplets at the surface of composites after conditioning at room temperature (RT) and after conditioning in the weathering chamber (WCh): (**A**)—15sedA RT, (**B**)—15sedB RT, (**C**)—15sedA WCh, (**D**)—15sedB WCh.

**Figure 19 materials-15-06106-f019:**
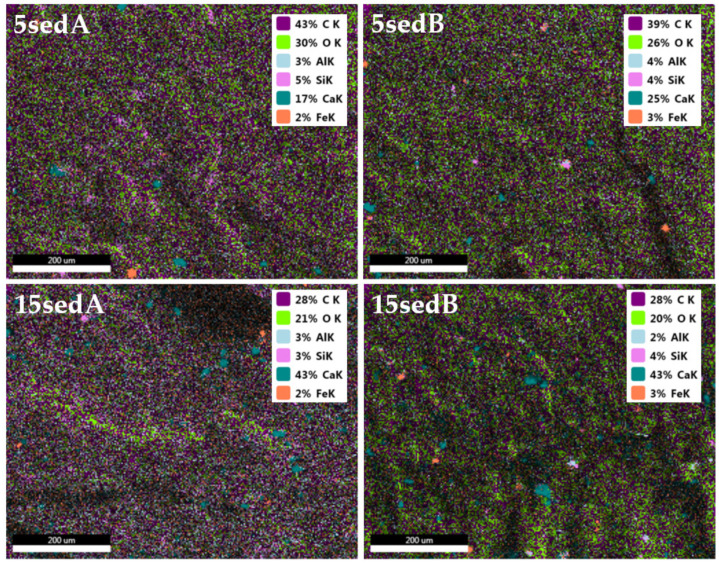
EDS of composites with the highest level of content.

**Figure 20 materials-15-06106-f020:**
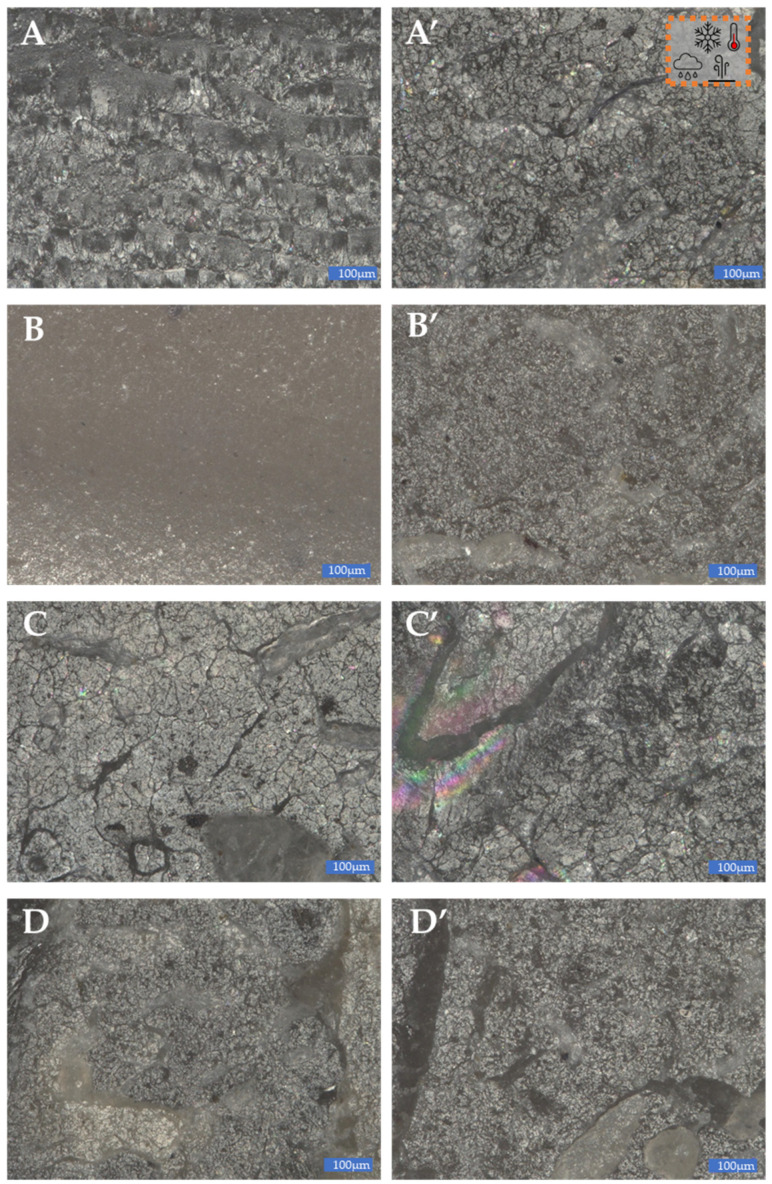
Optical microscope photos of composite fractures conditioned at room temperature (RT) and after conditioning in the weathering chamber (WCh): (**A**)–2.5sedA RT, (**A’**)–2.5sedA WCh, (**B**)–15sedA RT, (**B’**)–15sedA WCh, (**C**)–2.5sedB RT, (**C’**)–2.5sedB WCh, (**D**)–15sedB RT, (**D’**)–15sedB WCh.

**Figure 21 materials-15-06106-f021:**
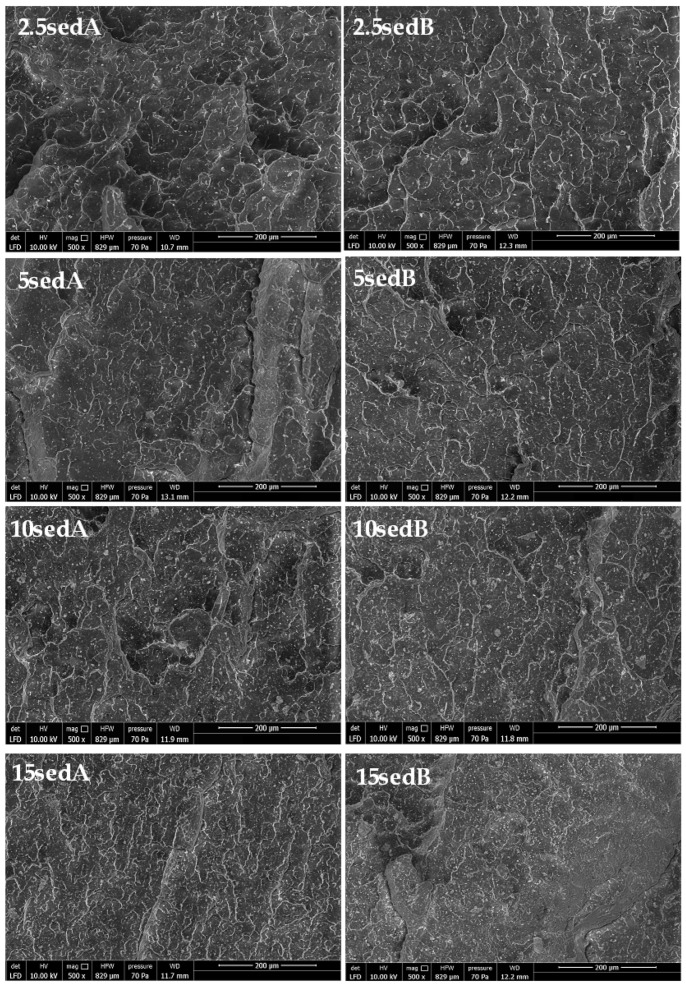
SEM images of PLA/CLS composite fractures.

**Table 1 materials-15-06106-t001:** Injection moulding parameters.

**Temperature [°C]**	**Nozzle**	**Zone 3**	**Zone 2**	**Zone 1**	**Traverse**
	190	195	200	185	40
**Holding pressure**		**t [s]**	0.0		9.0
		**P [bar]**	500		1500
**Closing Force [kN]**		**Holding Time [s]**	**Cooling Time [s]**		**Screw Diameter [mm]**
800		9	50		25

**Table 2 materials-15-06106-t002:** Final concentrations of the filler in the tested systems.

Full Name	Short Name
PLA2003D	PLA
PLA + 2.5% sediment 3–8 m < 40 µm	2.5sedA
PLA + 5% sediment 3–8 m < 40 µm	5sedA
PLA + 10% sediment 3–8 m < 40 µm	10sedA
PLA + 15% sediment 3–8 m < 40 µm	15sedA
PLA + 2.5% sediment 8–12 m < 40 µm	2.5sedB
PLA + 5% sediment 8–12 m < 40 µm	5sedB
PLA + 10% sediment 8–12 m < 40 µm	10sedB
PLA + 15% sediment 8–12 m < 40 µm	15sedB

**Table 3 materials-15-06106-t003:** Summary of the wavelengths in the FTIR analysis of bottom sediment samples.

Wavenumber [cm^−1^] sedA	Wavenumber [cm^−1^] sedB
1411	1411
1088	1088, 1029
864	864
705	705

**Table 4 materials-15-06106-t004:** Thermal decomposition of raw deposits.

	1st Decomposition Stage			2nd Decomposition Stage Residual Mass [%]			
Sample Name	Onset 1 [°C]	Peak 1 [°C]	Mass Change 1 [%]	Onset 2 [°C]	Peak 2 [°C]	Mass Change 2 [%]	Residual Mass [%]
sedA	254	302	5.6	681	731	45.5	48.9
sedB	250	304	5.6	686	736	44.9	49.4

## Data Availability

Not applicable.

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
