# Peer review of "Carbonate Lake Sediments in the Plastics Processing-Preliminary Polylactide Composite Case Study: Mechanical and Structural Properties"

_materials, 2022, doi:10.3390/ma15176106_

Round 1

Reviewer 1 Report

In this study, the influence of carbon lake sediments on the mechanical and structural properties of polylactide matrix composites was investigated. I think this is the manuscript is good. But the manuscript need a minor revise before publish.

(1) I think the cm-1 in the manuscript is wrong.

(2) Line 175, what is mean of the “i”?

(3) Figure2. I think the authors should unify the names of the samples. (sediment B, SED B)

(4) I think the introduction part of this manuscript was bad. Because there is no clear explanation of the significance of this research and the research progress in this field.

(5) Moderate English and format changes required.

Reviewer 2 Report

In this paper, Borokowski et al. examined the impact of sediment on the physical, mechanical and rheological properties of the PLA composites loaded with carbon sediments. The results are interesting. I am OK with publication of this manuscript after considering the following comments:

Authors are highly recommended to report the whole rheological data measured by rotational rheometry. Although MFR results provide interesting information, they are unable to provide good understanding about the primary rheological properties. Including elasticity and shear viscosity, of the composites.  If possible, please add storage and loss moduli data as well.

English needs further checking. Please also update your references.

Reviewer 3 Report

The manuscript by Borkowski and co-authors presents an interesting study on the improvement/modification of PLA properties by the addition of lake sediments. The results demonstrate different effects on elongation %, although statistical information is missing for all the data presented.

General comments

- Introduction section must be improved. The main goals of the research and its outcomes are not clear.

- Materials and method section: please add basic information (model, brand, supplier, city, country) of the used equipment. Add references for the methods used if adequate.

- Please add statistical information for the data on Figure 5, 7, 8, 9, 10, 11, 12, 13, 14 and 15. Also, it would be of great value for the discussion to include statistically significant difference (or not) on the modified composites properties.

- Can the authors suggest potential uses for the resulting composites based on its properties?

Specific comments

L 46. Does the nature/composition of the sediments also chance according to dept?

L 54. Please specify what PLA means on the first time it appears on text.

L 56. PLA can be produced via chemical synthesis or microbial fermentation. Please check https://doi.org/10.1111/jam.14290 and other sources. Add references accordingly.

L 70. Please specify PLA/CLS abbreviation.

L 71. DLS = Dinamic light scaterring, please corrected throughout the text.

L 148. Section title should be Results and Discussion.

L 232. Figure 6 quality must be improved.

L 236. Did the authors consider performing Differential scanning calorimetry to measure improvements on Tm, Tg etc. of the different composites?

Round 2

Reviewer 3 Report

The revised manuscript is suitable for publication in Materials.